# A 4K-Capable FPGA Implementation of Single Image Haze Removal Using Hazy Particle Maps

**Dat Ngo**[ID]**, Gi-Dong Lee and Bongsoon Kang ***

Department of Electronic Engineering, Dong-A University, Busan 49315, Korea
* Correspondence: bongsoon@dau.ac.kr; Tel.: +82-51-200-7703

**Abstract:** This paper presents a fast and compact hardware implementation using an efficient haze removal algorithm. The algorithm employs a modified hybrid median filter to estimate the hazy particle map, which is subsequently subtracted from the hazy image to recover the haze-free image. Adaptive tone remapping is also used to improve the narrow dynamic range due to haze removal. The computation error of the proposed hardware architecture is minimized compared with the floating-point algorithm. To ensure real-time hardware operation, the proposed architecture utilizes the modified hybrid median filter using the well-known Batcher's parallel sort. Hardware verification confirmed that high-resolution video standards were processed in real time for haze removal.

**Keywords:** hardware implementation; haze removal; hazy particle map; modified hybrid median filter; Batcher's parallel sort

## 1. Introduction

With the development of self-driving vehicles and intelligent monitoring systems, there is a growing demand for haze removal algorithms [1–3]. Loss of contrast and color fidelity of observed images due to hazy weather can strongly interfere with these applications. Therefore, several image processing algorithms are needed to improve the visibility of observed images. Some of them include haze removal, noise reduction, and wide dynamic range expansion.

The haze removal algorithms can be roughly divided into two main categories: multiple image processing [4,5] and single image processing [6–17]. Although multiple image processing is generally superior in performance, it is not attractive due to the high complexity of frame storage memories. Therefore, the recent focus has shifted to single image processing. He et al. proposed the dark channel prior (DCP) [6], which is used to estimate the transmission map based on the assumption that a few pixels with at least one very dark channel are present. However, block artifacts occurred in the estimated transmission due to the pre-assumed constant transmission in local patches. Therefore, the computationally intensive soft matting [18] was used additionally to refine the transmission, resulting in a very slow processing speed of the DCP. Tarel et al. proposed a faster haze removal method with a standard median filter instead of the time-consuming soft matting approach [7]. Thereby, the processing time was shortened, and the haze was reduced. However, this method resulted in halo artifacts in the corner (or line) areas with depth discontinuities caused by the large median filter for the estimation of the whiteness of the observed image. Nishino et al. proposed a Bayesian probability-based haze removal method utilizing a factorial Markov random field to model the dependence between scene albedo and depth from a single hazy image [8]. The algorithm effectively reduced the haze and color artifacts. However, it was also a time-consuming method requiring manual parametrization by depth prior [3,8]. Zhu et al. found that the difference between the brightness and saturation in a hazy image was generally correlated with the increasing haze concentration along with the scene depth [9]. Therefore, a linear model called color attenuation prior (CAP) was set up to estimate the

scene depth. Using the linear model and the guided image filter [19], the processing time and the haze in the observed images were reduced. However, this approach failed to handle grayscale images and dark regions in the images adequately. Bui et al. proposed a color ellipsoid prior (CEP) [15], which was theoretically similar to DCP as proved by Gibson et al. [16]. They also proposed using the fuzzy segmentation process in lieu of soft matting and derived an efficient formula for estimating and refining the transmission map concurrently. This could reduce haze removal time. However, experiments with provided code and parameters have shown that CEP tends to excessively dehaze the input images, causing visible color artifacts in the results. Liu et al., in contrast, proposed a unified strategy to estimate the transmission map and scene radiance simultaneously using total variation-based regularization [17]. Nevertheless, this iterative method required the atmospheric light and transmission map estimated by DCP in its initialization stage. Therefore, it was not computationally friendly. Most importantly, all the algorithms mentioned above require multiple frame memories, which greatly increase the hardware complexity at a high cost in the hardware implementation phase. Another approach proposed by Kim et al. removed haze from hazy images using hazy particle maps, estimated via modified hybrid median filter (mHMF) [11]. It used fewer frame storage memories in the hardware design, and its algorithmic complexity was solely a linear function of input image pixels, as will be discussed in Section 2.

In real-time image processing systems, a fast and compact solution for these algorithms is always essential. Park et al. proposed a fast implementation of DCP by lightening the complexity of the atmospheric light estimation and exploiting the high similarity of successive video frames [12]. In contrast, Salazar-Colores et al. shortened the processing time of DCP by utilizing morphological reconstruction instead of soft matting to refine the transmission map [13]. According to the provided experimental results, Park et al.'s design was able to process a $320 \times 240$ image in 0.228 s (i.e., $1/0.228 \approx 4.39$ frames per second (fps)), while Salazar-Colores et al.'s design required 0.14 s to handle a $1920 \times 1080$ image (i.e., $1/0.14 \approx 7.14$ fps). These faster versions of DCP are still inappropriate for real-time applications requiring a processing speed of at least 25 fps. Similarly, Table 1 shows the system complexity based on the CPU times using He et al.'s, Tarel et al.'s, Nishino et al.'s, Zhu et al.'s, Bui et al.'s, and Kim et al.'s algorithms [6–9,11,15]. Nishino et al.'s algorithm was programmed via Anaconda, while the other five algorithms were programmed via MATLAB R2019a. They were all tested on a Core i7-6700 CPU (3.4 GHz) with 16GB RAM. For the smallest $320 \times 240$ image, the fastest Bui et al.'s algorithm was only able to handle up to 20 fps (= 1/0.05). For the 4K standard of a $4096 \times 2160$ image, the speed of even the fastest Kim et al.'s algorithm was significantly declined to 0.09 fps ($\approx 1/11.09$), and the processing time of He et al.'s algorithm could not be measured due to insufficient RAM. This finding suggests that the real-time processing is completely impossible, and the memory capacity is a key factor in the hardware design for haze removal.

**Table 1.** Processing time in seconds of haze removal algorithms.

| Image Size | He [6] | Tarel [7] | Nishino [8] | Zhu [9] | Bui [15] | Kim [11] |
|---|---|---|---|---|---|---|
| $320 \times 240$ | 4.11 | 0.23 | 29.56 | 0.11 | 0.05 | 0.08 |
| $640 \times 480$ | 17.83 | 0.79 | 96.50 | 0.47 | 0.65 | 0.35 |
| $800 \times 600$ | 28.05 | 1.69 | 129.67 | 0.72 | 1.06 | 0.67 |
| $1024 \times 768$ | 46.82 | 2.18 | 173.86 | 1.18 | 1.74 | 0.96 |
| $1920 \times 1080$ | 143.86 | 4.27 | 583.18 | 3.14 | 4.76 | 2.25 |
| $4096 \times 2160$ | - | 25.76 | 2388.85 | 13.20 | 20.38 | 11.09 |

An alternative might be to implement these algorithms on a graphics processing unit (GPU) with GPU-accelerated libraries included in the Compute Unified Device Architecture toolkit. GPU can be a main means of realizing data-driven haze removal algorithms. For example, a cascaded convolutional neural network for dehazing proposed in Reference [14] was able to handle a $640 \times 480$ image in 0.1029 s (i.e., $1/0.1029 \approx 9.75$ fps). Unfortunately, this processing speed still does not meet the requirement of real-time processing. In addition, an in-depth study on both classical and deep learning haze removal

methods stated that the classical ones tend to generate results favored by human perception [20]. As well as this, researchers may be increasingly interested in designing the hardware using field-programmable gate arrays (FPGA) because of the high cost of the GPU platform in terms of both market price and energy consumption [21] and the inability to handle large image sizes in real time. Several hardware implementations until now including Shiau et al.'s and other designs [1,2,22] are only appropriate for small-size imaging applications because of their low throughput. Therefore, a strategy for processing a large size image in real time is highly required. This paper presents a 4K-capable hardware design using Kim et al.'s algorithm that meets the real-time processing criteria and does not require frame storage memory for consecutive images (or video frames) with high similarity. The proposed hardware architecture also utilizes the noted Batcher's parallel sorting algorithm [23] to realize the mHMF, consequently reducing the resource utilization and boosting the processing speed significantly.

The rest of the paper is organized as follows. Section 2 introduces the haze removal with hazy particle maps in detail. Section 3 describes the hardware architecture for the real-time processing of mHMF, a very important filter for the estimation of hazy particle maps. Section 4 presents the overall hardware design of the haze removal system and provides experimental results to demonstrate that the design can handle the 4K standard in real time. Section 5 concludes the paper.

## 2. Haze Removal with Hazy Particle Maps

The atmospheric scattering model, proposed by McCartney [24], is widely used to describe the formation of hazy images and is expressed as follows [6]:

$$I(x, y) = J(x, y)t(x, y) + A(1 - t(x, y)),  \tag{1}$$

where $(x, y)$ denotes the pixel locations in the hazy image $I$, the haze-free image $J$ as well as the transmission map $t$, and $A$ represents the global atmospheric light. Unfortunately, only the $I$ value is known in this equation. The goal of haze removal algorithms is to identify robust estimators of $A$ and $t$ to restore $J$ according to Equation (1). When white balance is performed correctly in an ideal environment, the haze looks completely white [7]. It means that the global atmospheric light $A$ can be set to one ($A = 1$). Then, the atmospheric scattering model is rewritten as follows:

$$\hat{I}(x, y) = J(x, y)t(x, y) + 1 - t(x, y),  \tag{2}$$

where $\hat{I}$ is the white-balanced image calculated using Equation (3) and $wb(\cdot)$ refers to the operator denoting white-balance processing, which was described in Reference [7]. Kim et al. defined a new variable termed hazy particle maps, shown in Equation (4), representing the haze distribution of the hazy image. Since the haze distribution correlates with the scene depth, it can change abruptly along the corners or lines of observed objects. Hence, the mHMF that preserves corners and edges can be used to estimate the hazy particle map, which is assumed to show positive values of the minimum color channel at each pixel [11]. The following equations are adopted to obtain the scene radiance:

$$\hat{I}(x, y) = wb(I(x, y)),  \tag{3}$$

$$H(x, y) = 1 - t(x, y),  \tag{4}$$

$$\hat{H}(x, y) = k_h mhmf\left(min_{RGB}\left(\hat{I}(x, y)\right)\right),  \tag{5}$$

$$J(x, y) = \frac{\hat{I}(x, y) - \hat{H}(x, y)}{1 - \hat{H}(x, y)},  \tag{6}$$

where $\hat{H}$ stands for the estimated value of the hazy particle map $H$, $k_h$ represents a weight factor of $0 \leq k_h \leq 1$, $mhmf(\cdot)$ denotes the operator of mHMF, and $min_{RGB}(\cdot)$ is the operator choosing each pixel's

minimum value. The last step is to perform adaptive tone remapping to expand the dynamic range of the recovered scene radiance $J$ [25]:

$$EL(x,y) = L(x,y) + G_L(x,y)W_L(x,y), \tag{7}$$

$$EC(x,y) = C(x,y) + G_C(x,y)W_C(x,y) + 128, \tag{8}$$

where $EL$ denotes the expanded luminance, $L$ is the input luminance, $G_L$ represents the luminance gain, $W_L$ stands for the adaptive luminance weight, $EC$ refers to the enhanced color, $C$ is the input color, $G_C$ is the color gain, and $W_C$ is the adaptive color weight. Readers can refer to Reference [25] for a detailed description on the calculation of luminance/color gains and adaptive weights.

A block diagram of the hazy particle map-based algorithm is shown in Figure 1. It comprises four consecutive steps: white balance, hazy particle map estimation, scene radiance recovery and adaptive tone remapping. $EJ$ is the dehazed output image composed of $EL$ and $EC$ with expanded dynamic range. It can be seen from the block diagram in Figure 1 and Equations (3)–(8) that the haze removal using hazy particle maps is a fact and efficient algorithm, since the most complicated operation is solely the mHMF in the white balance and hazy particle map estimation blocks. Other computations are simple and then can be done in constant time. For an image of size H × W and the mHMF of window size N × N, the complexity of this approach is O(H × W × N² × logN) when using brute-force implementation of mHMF, where logN refers to the binary logarithm of N. Fortunately, the mHMF operation can be done in constant time by virtue of the constant time median filter provided by Reference [26]. In this context, constant time means that the filtering operation is independent of the window size. Therefore, the algorithmic complexity of Kim et al.'s algorithm is a linear function of the number of image pixels O (H × W). This could be verified by the results of processing time analysis in Table 1.

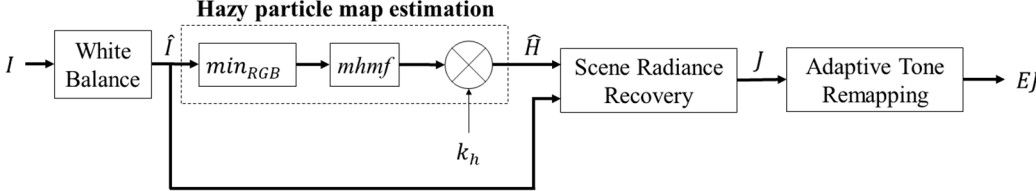

**Figure 1.** Block diagram of the haze removal using hazy particle maps.

In Reference [11], authors provided a comprehensive assessment of the dehazing performance with three benchmarking methods [6,7,9] on both real hazy image dataset [27] and synthetic dataset [28]. In order to replicate their work, we carry out a thorough evaluation on three new image datasets, including D-HAZY [29], O-HAZE [30] and I-HAZE [31]. D-HAZY is a synthetic dataset consisting of over 1400 images of real scenes and corresponding depth maps captured by a Microsoft Kinect camera. The atmospheric scattering model is then employed to synthesize hazy images with two assumptions: (i) the atmospheric light is uniform and (ii) the haze is homogeneous. O-HAZE and I-HAZE, on the other hand, are real datasets comprising pairs of hazy and clear images of outdoor and indoor scenes, respectively. To evaluate performance of haze removal algorithms, Mean Squared Error (MSE), Structural Similarity (SSIM) [32], Tone-Mapped image Quality Index (TMQI) [33] and Fog Aware Density Evaluator (FADE) [34] are used. SSIM and TMQI take on values within the range [0,1], in which higher values are better. In contrast, FADE is used to assess the haze density in a scene, a smaller FADE usually means that haze is more efficiently removed.

Table 2 displays the average MSE, SSIM, TMQI and FADE results on D-HAZY, O-HAZE and I-HAZE datasets, respectively. The best result is marked bold. On D-HAZY dataset, He et al.'s algorithm is best performing method under the MSE and SSIM metrics, while Kim et al.'s algorithm exhibits best performance under the TMQI and FADE metrics. On O-HAZE and I-HAZE datasets, the dehazing methods proposed by He et al. and Kim et al. show the best results in general, respectively.

Interestingly, Kim et al.'s method possesses the best TMQI score over all three datasets due largely to the adaptive tone remapping. This post-processing step efficiently resolves the narrow dynamic range of the haze removal process.

**Table 2.** Average Mean Squared Error (MSE), Structural Similarity (SSIM), Tone-Mapped image Quality Index (TMQI), and Fog Aware Density Evaluator (FADE) results on three datasets.

| Dataset | Method | MSE | SSIM | TMQI | FADE |
|---------|--------|------|------|------|------|
| D-HAZY | He | **0.0303** | **0.8348** | 0.8631 | 0.7422 |
| | Tarel | 0.0611 | 0.7475 | 0.8000 | 0.9504 |
| | Zhu | 0.0481 | 0.7984 | 0.8206 | 0.9745 |
| | Kim | 0.0373 | 0.7672 | **0.8782** | **0.5054** |
| O-HAZE | He | **0.0200** | **0.7709** | 0.8403 | **0.3719** |
| | Tarel | 0.0283 | 0.7263 | 0.8416 | 0.4013 |
| | Zhu | 0.0274 | 0.6647 | 0.8118 | 0.6531 |
| | Kim | 0.0346 | 0.7255 | **0.8921** | 0.4038 |
| I-HAZE | He | 0.0535 | 0.6580 | 0.7319 | 0.8328 |
| | Tarel | 0.0318 | 0.7200 | 0.7740 | **0.8053** |
| | Zhu | 0.0359 | 0.6864 | 0.7512 | 1.0532 |
| | Kim | **0.0237** | **0.7672** | **0.7978** | 0.9632 |

Hazy scenes with corresponding dehazed images are presented in Figure 2 to visually assess the dehazing performance of four methods. In this case, it is evident that dehazing approaches proposed by He et al. and Kim et al. share the top performance. This is consistent with the quantitative results in Table 2. He et al.'s method accidentally enlarges the sun after dehazing in the first row and suffers from background noise in the sky of the second row. Tarel et al.'s method introduces halo artifacts at small edges like the tree twigs or the pointed roofs in the first two rows. Zhu et al.'s method produces too dark image because of large haze removal in the first and third rows. Only Kim et al.'s method shows visually-good results for three input hazy images.

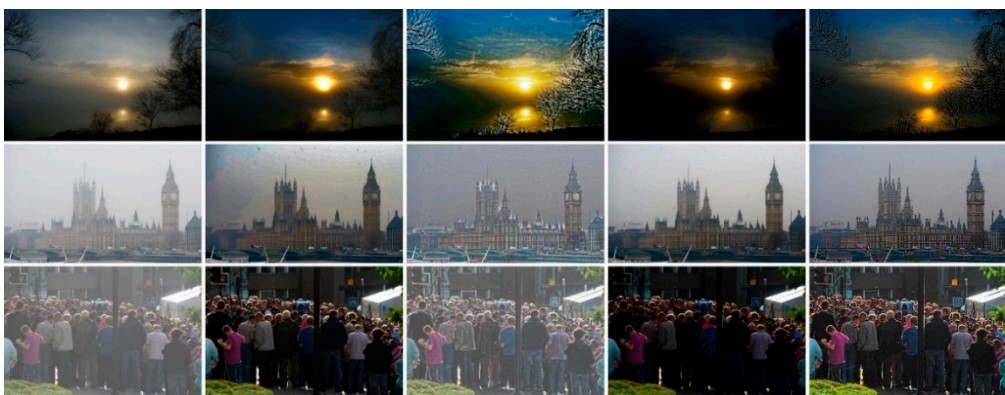

**Figure 2.** Qualitative evaluation of haze removal algorithms: (from left to right) input hazy image and dehazed images of He et al., Tarel et al., Zhu et al., and Kim et al.

Based on this quantitative and qualitative evaluations, as well as the comparison of running time for various image sizes in Table 1, we decided to implement Kim et al.'s algorithm by the means of FPGA to meet the real-time processing requirement. Consequently, our proposed FPGA design is highly appropriate for smart surveillance system or self-driving vehicles. The first step in the hardware design procedure is to convert the floating-point algorithm into its fixed-point version. The brief introduction to this conversion is in order. A real number can be represented by a 32-bit, 64-bit, or 128-bit floating-point value *FL* with high precision (IEEE Standard 754-2019 [35]) or by a reduced-precision *N*-bit fixed-point value *FI*, where *N* is greatly smaller than 32. Floating-point number representation is

widely used in software for algorithm design and simulation, while fixed-point number representation is mainly used in hardware design. An *N*-bit fixed-point representation is actually an *N*-bit binary number with an 'implied binary point' separating the number into *FC*-bit fraction and $(N - FC)$-bit integer parts, analogous to the decimal point between the fraction and integer digits of an ordinary decimal number. By converting a number from its floating-point representation to its fixed-point representation using Equation (9), reduced hardware utilization and increased processing speed can be achieved at the cost of losing arithmetic accuracy. Accordingly, hardware designers have to decide the number of fraction bits for every signal in the design so that the propagated loss of accuracy at the output is within a predetermined limit. In addition, they also allocate the number of integer bits to adequately accommodate the signal range. Our hardware design can achieve the computation error of less than ±0.5 Least Significant Bit (LSB) compared with the floating-point algorithm. This computation error is calculated using Equation (10).

$$FI = round\left(FL \times 2^{FC}\right), \tag{9}$$

$$E = FL_{EJ} - \frac{FI_{EJ}}{2^{FC_{EJ}}} \tag{10}$$

where $E$ is the computation error, $FL_{EJ}$ denotes the floating-point value, and $FI_{EJ}$ represents the fixed-point value of $FL_{EJ}$ with $FC_{EJ}$ fraction bits at the final output *EJ* in Figure 1.

The visual evaluation in Figure 2 shows that Kim et al.'s algorithm resolves completely halo artifacts that occur in Tarel et al.'s approach. This is achieved by employing the mHMF, which outperforms the standard median filter used in Tarel et al.'s method to estimate the haze distribution. Hence, it is worthwhile to design a compact hardware architecture for the mHMF that can handle large size images in real time.

## 3. Hardware Architecture for Modified Hybrid Median Filter

The median filter is an effective filter that preserves edges in the areas with depth continuities [7]. As a result, the filter may leave halo artifacts in the areas of corners or lines with depth discontinuities. To overcome this drawback, a hybrid median filter (HMF) is needed to ensure corner areas with depth discontinuities [36]. Since outdoor scenes display a series of hazy distributions, it is better to use the median filter in areas with depth continuities and the HMF in areas with depth discontinuities. Therefore, a $7 \times 7$ mHMF was proposed according to the median of a square window instead of a window's center pixel [11], as depicted in Figure 3. The mHMF initially determines the medians of 140, 121, and 213 corresponding to the three windows and identifies the final median of 140. In the case of HMF, the final median of 213 was selected, which was significantly larger than that of mHMF since the center pixel was used instead of the median of the square window (213 instead of 140). Since this large value (213) highly likely refers to a noisy pixel, mHMF improves HMF performance to enhance the prediction of hazy particle maps.

Using the square window median instead of the center pixel enhances the performance in flat areas with depth continuities and corner areas with depth discontinuities, such as outdoor scenes. However, the hardware implementation is more burdensome since the hardware size of mHMF is substantially larger than that of HMF. The hardware design of each mHMF consists of three standard median filters corresponding to the three window types and a single three-input median filter for selecting the final median. In addition, a total of four mHMFs are needed: three mHMFs for white balancing of Equation (3) in RGB channels and a single mHMF for the estimation of the hazy particle maps of Equation (5). In general, the median filter can be implemented based on a sorting network [37] or without such a network [38]. A sorting network is appropriate for filtering with a small window size, but not for applications requiring a large window size.

| 192 | 245 | 178 | 220 | 64 | 234 | 14 |
|---|---|---|---|---|---|---|
| 70 | 87 | 227 | 65 | 157 | 73 | 135 |
| 173 | 149 | 245 | 220 | 121 | 193 | 199 |
| 215 | 57 | 140 | 213 | 90 | 215 | 238 |
| 41 | 192 | 35 | 237 | 212 | 97 | 33 |
| 30 | 65 | 38 | 89 | 149 | 145 | 145 |
| 127 | 129 | 66 | 50 | 140 | 19 | 120 |

(a)

| 192 | 245 | 178 | 220 | 64 | 234 | 14 |
|---|---|---|---|---|---|---|
| 70 | 87 | 227 | 65 | 157 | 73 | 135 |
| 173 | 149 | 245 | 220 | 121 | 193 | 199 |
| 215 | 57 | 140 | 213 | 90 | 215 | 238 |
| 41 | 192 | 35 | 237 | 212 | 97 | 33 |
| 30 | 65 | 38 | 89 | 149 | 145 | 145 |
| 127 | 129 | 66 | 50 | 140 | 19 | 120 |

(b)

| 192 | 245 | 178 | 220 | 64 | 234 | 14 |
|---|---|---|---|---|---|---|
| 70 | 87 | 227 | 65 | 157 | 73 | 135 |
| 173 | 149 | 245 | 220 | 121 | 193 | 199 |
| 215 | 57 | 140 | 213 | 90 | 215 | 238 |
| 41 | 192 | 35 | 237 | 212 | 97 | 33 |
| 30 | 65 | 38 | 89 | 149 | 145 | 145 |
| 127 | 129 | 66 | 50 | 140 | 19 | 120 |

(c)

**Figure 3.** Modified hybrid median filter: (**a**) square window, (**b**) diagonal window, and (**c**) cross window.

Figure 4 outlines the sorting procedure of a seven-input median filter. Figure 4a depicts the sorting procedure using the well-known odd-even sorting-network (OESN) [37], and Figure 4b displays a novel sorting procedure proposed in this paper based on Batcher's sorting network (BSN) [23]. The vertical line connecting the two inputs represents a compare-and-swap (CS) unit, which compares the two input values in the descending order. The CS units in the blue rectangular group operate concurrently in a single clock cycle. The proposed procedure based on BSN, consisting of six blue rectangular groups with 14 CS units, identifies the median in six clock cycles whereas OESN procedure with 18 CS units requires seven clock cycles. Notably, there are differences in the number of units and clock cycles: four and one, which significantly reduce the system complexity in hardware implementation.

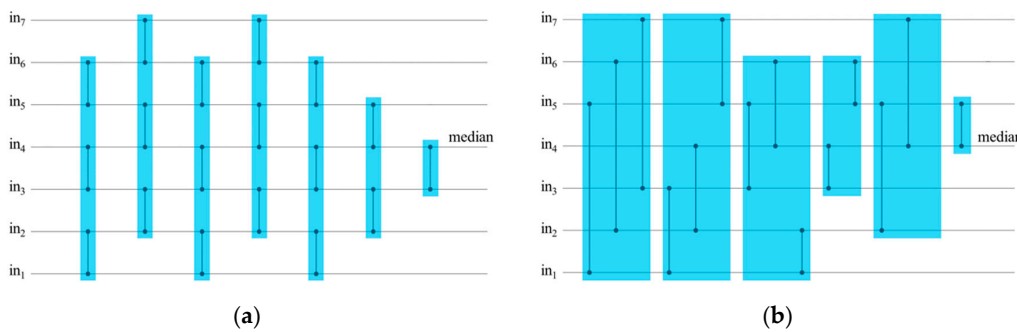

**Figure 4.** Sorting procedures of a seven-input median filter: (**a**) odd-even sorting network (OESN) and (**b**) the proposed method based on Batcher's sorting network (BSN).

The three hardware architectures of mHMF illustrated in Figure 5 have been designed and tested to compare the hardware complexity. To this end, a filter consisting of a $7 \times 7$ window filter and 10-bit data was used for mHMF. The three figures in Figure 5 display the hardware architectures implemented using OESN, cumulative histogram (CumHist) [38] and the proposed method, respectively. The sorting procedure in Figure 4a can be extended according to the $7 \times 7$ window size. The square window comprises 49 inputs, and the diagonal and cross windows contain 13 inputs. Therefore, the OSEN-based architecture depicted in Figure 5a consists of median filters with 49 and 13 inputs. A 3-input median filter is used to determine the final median at the output. Regarding the CumHist architecture in Figure 5b, the sizes of Read-Only Memories (ROMs) and a priority encoder depend on the number of bits in the input data and the window size of a median filter [38]. The 13-input OSEN-based median filter can also be used in diagonal and cross windows of the CumHist architecture. The hardware design used a $7 \times 7$ window filter and 10-bit data herein. Therefore, 14 1024 × 1024-bit ROMs, 1024 6-bit registers, and a 1024-to-10 priority encoder are required for the square window of the median filter. The sorting procedure with reduced CS units and clock cycles is used as the hardware architecture for the proposed method. Figure 5c shows the architecture, which also contains 49 and 13 inputs for

the median filters. The results of hardware implementation using the method proposed in this paper along with the other two methods are discussed in the next section.

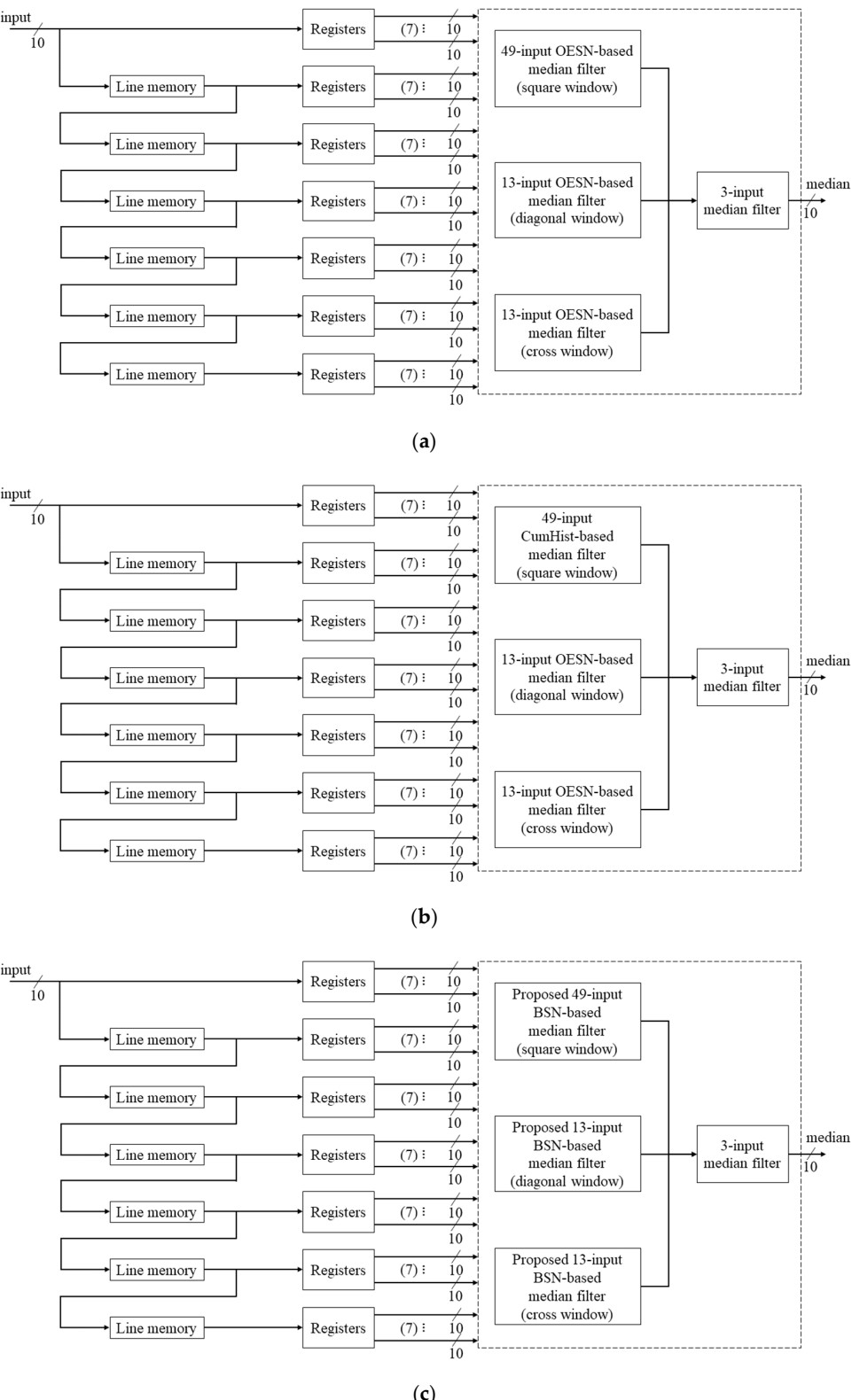

**Figure 5.** Hardware architectures of a 7 × 7 modified hybrid median filter (mHMF): (**a**) OESN, (**b**) CumHist, and (**c**) the proposed method based on BSN.

## 4. Hardware Implementation

In order to verify the complexity of the three hardware architectures shown in Figure 5, an mHMF filter was implemented in a system-on-a-chip (SoC) evaluation board (Xilinx Zynq-7000 SoC ZC706 Evaluation Kit) [39] using Verilog hardware description language. Table 3 summarizes the results of FPGA implementation. Registers and lookup tables (LUT) represent logical gate areas, and RAMB36E1 represents memory areas. The proposed method used 11,139 registers and 9745 LUTs, which were substantially smaller than those of OESN and CumHist. Therefore, the proposed method based on BSN was selected to implement the hardware design for haze removal using hazy particle maps.

**Table 3.** Field Programmable Gate Array (FPGA) implementation of a $7 \times 7$ mHMF.

| Slice Logic Utilization | OESN [37] | CumHist [38] | Proposed Method |
| --- | --- | --- | --- |
| Registers (#) | 21,870 | 42,264 | 11,139 |
| LUTs (#) | 22,756 | 42,060 | 9745 |
| RAMB36E1 | 6 | 6 | 6 |

Figure 6 shows a detailed block diagram of a 4K-capable FPGA implementation of haze removal using hazy particle maps. It comprises eight main processing blocks with 30 SPRAM memories. The input hazy images (or video frames) supplied to the system controller were provided by the Visual Studio platform running on the host computer for hardware experiments. The platform was programmed to extract frame data from the hazy image $I$ and feed them to the controller implemented on the SoC evaluation board. The controller reads and stores the image data using double-buffering. The hazy image was processed to estimate the hazy particle map $\hat{H}$ in Equation (5) after white balancing, and the result was used to recover scene radiance $J$ in Equation (6). The macro block provides mathematical functions such as dividers and square rooters. The blocks of RGB-to-YCbCr422 and YCbCr422-to-RGB convert the format of color component signals. The recovered scene radiance was supplied to the adaptive tone remapping to restore video visibility of the final output image $EJ$ comprising $EL$ and $EC$ in Equations (7) and (8), respectively. This output image was then fed to the platform to analyze the performance of the proposed design on the host computer. The two memories $1024 \times 32$-bit SPRAMs were used to construct the histogram required for the adaptive tone remapping [25]. Other memories were used as line buffers to filter operations in white balancing and hazy particle map estimation. The proposed architecture does not use any of SPRAMs as frame storage memories.

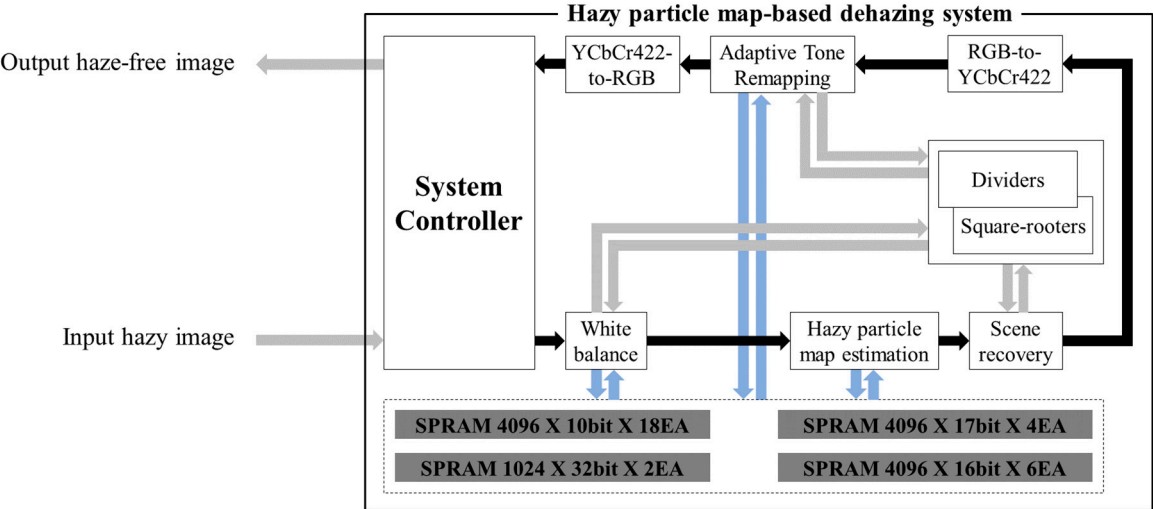

**Figure 6.** Hardware architecture of the haze removal using hazy particle maps.

The SoC evaluation board integrates an FPGA, application process units, and numerous subsystems. The FPGA implementing the proposed design for the evaluation of real-time performance is shown in the bottom third of Figure 7. The top third and the middle third of Figure 7 represent the Visual Studio platform, which provides input images to and displays the results from the evaluation board. In the top third, the left panel shows an input hazy image, and the right panel shows the dehazed image obtained using the hardware design proposed in this paper. In the middle third, the left platform control allows users to select the input data (e.g., a still image, a video stream from a webcam or a file on the host computer) and to save the results. The right algorithm control allows users to change the parameters provided to the hardware implemented on the SoC evaluation board. This Visual Studio platform was used to evaluate the performance of the proposed design.

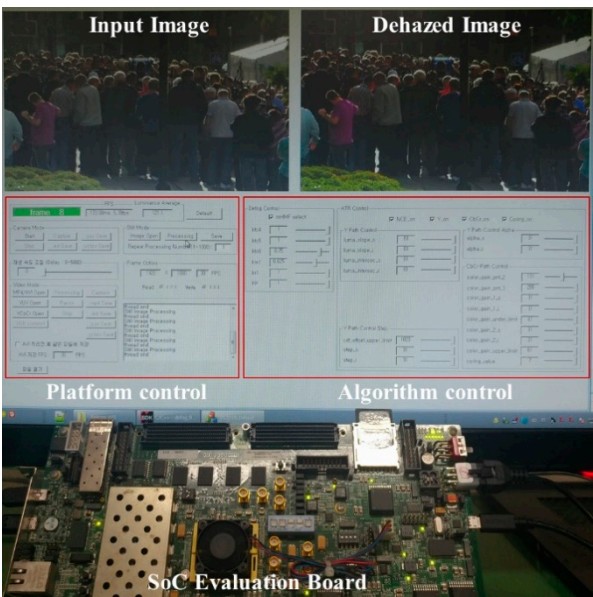

**Figure 7.** Verification using a SoC evaluation board.

Table 4 summarizes the results of FPGA implementation. The entire haze removal system utilized approximately 16.21% of registers and 25.92% of LUTs. The memory areas used for RAMB36E1 were 85, respectively, which constituted 15.60% of the total available memories. The maximum operating frequency of the implemented hardware was 236.29 MHz. Considering additional horizontal and vertical blank periods in hardware implementation, the maximum processing speed (MPS) in fps can be calculated easily using the following equation:

$$\text{MPS} = \frac{f_{max}}{(W + HB)(H + VB)},$$

(11)

where $f_{max}$ is the maximum operating frequency, $W$ denotes the width of the input image, $HB$ represents the horizontal blank, $H$ refers to the height of the input image, and $VB$ indicates the vertical blank. In this study, the proposed design required a clock cycle and a horizontal line for each horizontal and vertical blank period.

**Table 4.** FPGA implementation of the proposed method.

| Slice Logic Utilization | Available | Used | Utilization |
|---|---|---|---|
| Registers (#) | 437,200 | 70,864 | 16.21% |
| LUTs (#) | 218,600 | 56,664 | 25.92% |
| RAMB36E1 | 545 | 85 | 15.60% |
| Minimum Period (ns) | | 4.23 | |
| Maximum Frequency (MHz) | | 236.29 | |

Table 5 shows the processing speeds for various video standards. The number of clock cycles required to process the DCI 4K video standard of a 4096 × 2160 image was 8,853,617 (= 4097 × 2161), suggesting that the proposed haze removal system can handle up to 26.7 fps (= $236.29 \times 10^6/8,853,617$) and meets the real-time processing criteria of handling at least 25 fps. This result demonstrates that the proposed method can be applied to real-time applications using up to 4K video standards.

**Table 5.** Processing speeds for various video standards.

| Video Standard | | Image Size | Required Clock Cycles (#) | Processing Speed (fps) |
|---|---|---|---|---|
| Full HD (FHD) | | 1920 × 1080 | 2,076,601 | 113.8 |
| Quad HD | | 2560 × 1440 | 3,690,401 | 64.0 |
| 4K | UW4K | 3840 × 1600 | 6,149,441 | 38.4 |
| | UHD TV | 3840 × 2160 | 8,300,401 | 28.5 |
| | DCI 4K | 4096 × 2160 | 8,853,617 | 26.7 |

The comparison with other hardware designs is also shown in Table 6. Since Shiau et al. and Zhang et al. did not present the implementation results thoroughly as Park et al. did, their data of Registers, LUTs, DSPs (Digital Signal Processing blocks), and Memory are listed as NA (Not Available). In Reference [1], Shiau et al. claimed that their design can achieve 200 MHz using TSMC 0.13-μm technology. However, when they implemented it on Altera Stratix EP1S10F780C6 FPGA platform, the maximum frequency is only 58.43 MHz, leading to the fact that it is solely fast enough to process a video resolution of FHD at 28.1 fps in real time. Park et al. proposed a hardware architecture for haze removal at fixed frame sizes of 320 × 240, 640 × 480, and 800 × 600. Accordingly, even though their design achieved the maximum frequency of 88.70 MHz, it can only process the maximum video resolution of Super VGA (SVGA). Adding to that, it requires 32,000 ALMs (Adaptive Logic Modules), which comprises two adaptive lookup tables and then can be inferred to 64,000 LUTs. Except the number of registers, more LUTs, DSPs, and memory are needed to implement a design that cannot be compared to our proposed design in terms of processing speed and resource utilization. Finally, the design of Zhang et al. can be considered as an improvement upon Shiau et al.'s under the processing speed metric. Their design is capable of processing a Quad HD video resolution at 31.4 fps in real time. All the calculations of processing speed for designs of Shiau et al. and Zhang et al. are based on the assumption that they also require a clock cycle and a horizontal line for each horizontal and vertical blank period. In fact, their requirement for horizontal and vertical blank period may be different, and the processing speed may decrease. Therefore, it is evident from Table 6 that our proposed hardware architecture is superior to other benchmarking designs in term of processing speeds in References [1,2,22] and requires less resources than the design in Reference [2].

**Table 6.** Comparison of FPGA synthesized results.

| Hardware Utilization | Shiau et al. [1] | Park et al. [2] | Zhang et al. [22] | Proposed Method |
|---|---|---|---|---|
| Registers (#) | NA | 53,400 | NA | 70,864 |
| LUTs (#) | NA | 64,000 | NA | 56,664 |
| DSPs (#) | NA | 42 | NA | 0 |
| Memory (Mbits) | NA | 3.2 | NA | 1.5 |
| Maximum Frequency (MHz) | 58.43 | 88.70 | 116.00 | 236.29 |
| Maximum Video Resolution | FHD | SVGA | Quad HD | DCI 4K |

As mentioned in Section 2, the computation error of the proposed hardware architecture is within ±0.5 LSB compared with the floating-point algorithm. This may cause a possible issue of slight color shift, as illustrated in Figure 8. The color around the train headlights is slightly stronger due to the limited bit width of every signal in the hardware implementation. Except for this kind of minor shortcoming, the results of two methods shown in Figure 8 are virtually identical. The results of the implemented hardware in Table 7 are very similar to those of Kim et al.'s algorithm. Small differences in the results were caused by the limited bit widths to achieve compact and fast hardware designs. In the floating-point algorithm, $k_h$ in Equation (5) could be fine-tuned to get the best results, as shown in Table 2. However, when converting to the fixed-point representation for hardware implementation, we allocated four bits to the fraction of $k_h$. As a result, $k_h$ could only be coarse-tuned in the hardware design at intervals of 0.0625 (=$1/2^4$). Accordingly, even when $k_h = 0.9$ gives the best result when tuning the floating-point algorithm, we can only select $k_h = 0.875$ or 0.9375 in the implemented hardware. This leads to a slight difference, as shown in Figure 8 and Table 7.

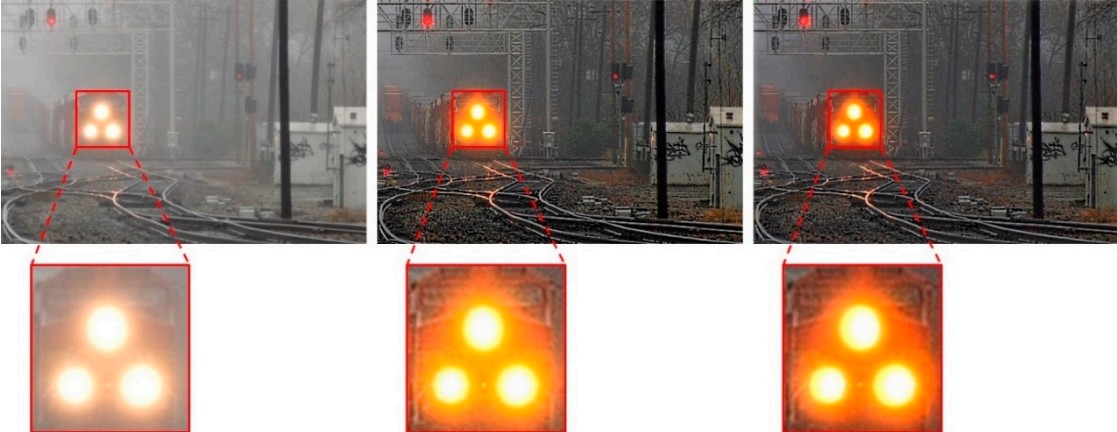

**Figure 8.** From left to right: input hazy image, dehazed images of the floating-point algorithm and our proposed hardware design, and their corresponding zoomed-up regions.

**Table 7.** Average MSE, SSIM, TMQI and FADE results on three datasets.

| Dataset | Method | MSE | SSIM | TMQI | FADE |
|---|---|---|---|---|---|
| D-HAZY | Ours | 0.0430 | 0.7738 | 0.8615 | 0.5143 |
| | Kim | 0.0373 | 0.7672 | 0.8782 | 0.5054 |
| O-HAZE | Ours | 0.0220 | 0.7636 | 0.8917 | 0.4125 |
| | Kim | 0.0346 | 0.7255 | 0.8921 | 0.4038 |
| I-HAZE | Ours | 0.0246 | 0.7700 | 0.7883 | 0.9919 |
| | Kim | 0.0237 | 0.7672 | 0.7978 | 0.9632 |

## 5. Conclusions

A fast and efficient hardware architecture for haze removal was presented in this paper. The hardware was compact due to the novel Batcher's sort-based modified hybrid median filter implementation and provided high throughput for real-time video processing based on pipeline architecture. In order to guarantee the high performance of the proposed architecture, the hardware was also designed to minimize the computation error to less than ±0.5 LSB compared with the floating-point algorithm. The maximum operating frequency of the designed hardware was 236.29 MHz, which was fast enough to handle 4K video standards in real time at 26.7 fps. Finally, the designed hardware was verified on the SoC board.

**Author Contributions:** D.N. designed and experimented the hardware; G.-D.L. and B.K. conceived the idea of the paper; B.K. wrote the manuscript with the editing of all authors and is the director of this work.

**Funding:** This research was supported by research funds from Dong-A University, Busan, South Korea.

**Conflicts of Interest:** The authors declare no conflict of interest.

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
