# Peer review of "A 4K-Capable FPGA Implementation of Single Image Haze Removal Using Hazy Particle Maps"

_applsci, doi:10.3390/app9173443_

Round 1

Reviewer 1 Report

The authors claim to present a fast and compact hardware implementation using a haze-based removal algorithm. The system uses a modified hybrid median filter to estimate the hazy particle map which is subsequently subtracted from the source image in order to recover the haze-free map. Adaptive tone remapping is also used to improve the narrow dynamic range due to haze removal. The authors allege that the computation error of the proposed hardware architecture is minimized compared with the floating-point algorithm. The system has been designed to meet with real-time constraints. Hardware verification confirmed that high-resolution video standards were processed in real-time for haze removal

This paper deals with an interesting topic, nevertheless, it can not be accepted in a journal such as Applied Science in its present form.

-1- First major problem is concerning the completeness of the work, the authors must enhance the introductory part citing previous work regarding hazing system and methods such as:

[a]Y. Park et al.  "Fast Execution Schemes for Dark-Channel-Prior-Based Outdoor Video Dehazing," in IEEE Access, vol. 6, pp. 10003-10014, 2018.

[b]Salazar-Colores, S. et al. (2019). A Fast Image Dehazing Algorithm Using Morphological Reconstruction. IEEE Transactions on Image Processing, 28(5), 2357-2366.
[c]B. Li et al., "Benchmarking Single-Image Dehazing and Beyond," in IEEE Transactions on Image Processing, vol. 28, no. 1, pp. 492-505, Jan. 2019.
[d]Y. Liu, J. Shang, L. Pan, A. Wang, and M. Wang, "A Unified Variational Model for Single Image Dehazing," in IEEE Access, vol. 7, pp. 15722-15736, 2019.
[e]T. M. Bui and W. Kim, "Single Image Dehazing Using Color Ellipsoid Prior," in IEEE Transactions on Image Processing, vol. 27, no. 2, pp. 999-1009, Feb. 2018.
[f]C. Li, J. Guo, F. Porikli, H. Fu and Y. Pang, "A Cascaded Convolutional Neural Network for Single Image Dehazing," in IEEE Access, vol. 6, pp. 24877-24887, 2018.

-2- Second major problem lies in the conversion between floating vs integer format. A deep analysis is expected. Besides refs to other computer vision works that deal with this topic must be added such as:

[g] Botella, G. et al. (2012). Quantization analysis and enhancement of a VLSI gradient-based motion estimation architecture. Digital Signal Processing, 22(6), 1174-1187.

In addition, the relation error vs bit-width taken should be presented here.

-3- The third major problem is regarding the methodology...
How can I replicate this methodology? What is the computational complexity order of the algorithm proposed? What conclusion should I extract from this?
The authors must perform a complete analysis against every parameter that comes up in the system. Otherwise, this paper lacks generality. In the way that this algorithm is explained is difficult to reproduce the experiment.

Reviewer 2 Report

The paper proposes a new implementation of an Haze removal algorithm, using FPGA.
The authors presented a compreensive state-of-the-art, both on the best algorithm to implement and on related implementations.

The tecnical description of the algorithm and its implementation, is suited for an international journal.
The authors validated the proposed method correctly, with 26.7 fps which is suitable for the declared objectives on DCI 4K.

The paper is missing a discussion on possible drawbacks on the proposed implementation. Does the specific implementation resulted in any decay of the base algorithms performance? i.e., image quality. The paper is missing a discussion on image quality issues that may occur, or not, on the proposed method. Descriptors on this should also be presented and discussed.

Reviewer 3 Report

Figures 1, 2 and 4; should be moved after the next paragraphs

Fig.3; missing the reference in the text

Fig. 4b; to move to left (cut off word "median")

Fig. 7; Could be split into two figures or delete down part (board). Description of the figure in the text does not correspond with the figure (line243)

Line246,247 - the sentence is unnecessary

Tab. 3; Title of the last column could be "Proposed method". There is missing "*" in the text

Tab. 4; whole word - frequency

Tab. 5; last column - speed; 4K UHD TV

Tab. 6; whole word - frequency, resolution; last column could be "Proposed method"

Conclusion - Lines 294,295; could be "The maximum processing speed ... was 26,7 fps ..."   

Round 2

Reviewer 1 Report

Thank you for the new additions, the authors have raised all my comments, thus, I support manuscript publication.